# Weighted importance sampling for off-policy learning with linear function approximation

**A. Rupam Mahmood, Hado van Hasselt, Richard S. Sutton**
Reinforcement Learning and Artificial Intelligence Laboratory
University of Alberta
Edmonton, Alberta, Canada T6G 1S2
{ashique,vanhasse,sutton}@cs.ualberta.ca

## Abstract

Importance sampling is an essential component of off-policy model-free reinforcement learning algorithms. However, its most effective variant, *weighted* importance sampling, does not carry over easily to function approximation and, because of this, it is not utilized in existing off-policy learning algorithms. In this paper, we take two steps toward bridging this gap. First, we show that weighted importance sampling can be viewed as a special case of weighting the error of individual training samples, and that this weighting has theoretical and empirical benefits similar to those of weighted importance sampling. Second, we show that these benefits extend to a new weighted-importance-sampling version of off-policy LSTD($\lambda$). We show empirically that our new WIS-LSTD($\lambda$) algorithm can result in much more rapid and reliable convergence than conventional off-policy LSTD($\lambda$) (Yu 2010, Bertsekas & Yu 2009).

## 1 Importance sampling and weighted importance sampling

Importance sampling (Kahn & Marshall 1953, Rubinstein 1981, Koller & Friedman 2009) is a well-known Monte Carlo technique for estimating an expectation under one distribution given samples from a different distribution. Consider that data samples $Y_k \in \mathbb{R}$ are generated i.i.d. from a sample distribution $l$, but we are interested in estimating the expected value of these samples, $v_g \doteq \mathbb{E}_g[Y_k]$, under a different distribution $g$. In importance sampling this is achieved simply by averaging the samples weighted by the ratio of their likelihoods $\rho_k \doteq \frac{g(Y_k)}{l(Y_k)}$, called the *importance-sampling ratio*. That is, $v_g$ is estimated as:

$$\tilde{v}_g \doteq \frac{\sum_{k=1}^n \rho_k Y_k}{n}. \tag{1}$$

This is an unbiased estimate because each of the samples it averages is unbiased:

$$\mathbb{E}_l[\rho_k Y_k] = \int_y l(y) \frac{g(y)}{l(y)} y \, \mathrm{d}y = \int_y g(y) y \, \mathrm{d}y = \mathbb{E}_g[Y_k] = v_g.$$

Unfortunately, this importance sampling estimate is often of unnecessarily high variance. To see how this can happen, consider a case in which the samples $Y_k$ are all nearly the same (under both distributions) but the importance-sampling ratios $\rho_k$ vary greatly from sample to sample. This should be an easy case because the samples are so similar for the two distributions, but importance sampling will average the $\rho_k Y_k$, which will be of high variance, and thus its estimates will also be of high variance. In fact, without further bounds on the importance-sampling ratios, $\tilde{v}_g$ may have infinite variance (Andradóttir et al. 1995, Robert & Casella 2004).

An important variation on importance sampling that often has much lower variance is *weighted importance sampling* (Rubinstein 1981, Koller & Friedman 2009). The weighted importance sampling

(WIS) estimate $v_g$ as a weighted average of the samples with importance-sampling ratios as weights:

$$\hat{v}_g \doteq \frac{\sum_{k=1}^{n} \rho_k Y_k}{\sum_{k=1}^{n} \rho_k} \, .$$

This estimate is biased, but consistent (asymptotically correct) and typically of much lower variance than the ordinary importance-sampling (OIS) estimate, as acknowledged by many authors (Hesterberg 1988, Casella & Robert 1998, Precup, Sutton & Singh 2000, Shelton 2001, Liu 2001, Koller & Friedman 2009). For example, in the problematic case sketched above (near constant $Y_k$, widely varying $\rho_k$) the variance of the WIS estimate will be related to the variance of $Y_k$. Note also that when the samples are bounded, the WIS estimate has bounded variance, because the estimate itself is bounded by the highest absolute value of $Y_k$, no matter how large the ratios $\rho_k$ are (Precup, Sutton & Dasgupta 2001).

Although WIS is the more successful importance sampling technique, it has not yet been extended to parametric function approximation. This is problematic for applications to off-policy reinforcement learning, in which function approximation is viewed as essential for large-scale applications to sequential decision problems with large state and action spaces. Here an important subproblem is the approximation of the value function—the expected sum of future discounted rewards as a function of state—for a designated target policy that may differ from that used to select actions. The existing methods for off-policy value-function approximation either use OIS (Maei & Sutton 2010, Yu 2010, Sutton et al. 2014, Geist & Scherrer 2014, Dann et al. 2014) or use WIS but are limited to the tabular or non-parametric case (Precup et al. 2000, Shelton 2001). How to extend WIS to parametric function approximation is important, but far from clear (as noted by Precup et al. 2001).

## 2  Importance sampling for linear function approximation

In this section, we take the first step toward bridging the gap between WIS and off-policy learning with function approximation. In a general supervised learning setting with linear function approximation, we develop and analyze two importance-sampling methods. Then we show that these two methods have theoretical properties similar to those of OIS and WIS. In the fully-representable case, one of the methods becomes equivalent to the OIS estimate and the other to the WIS estimate.

The key idea is that OIS and WIS can be seen as least-squares solutions to two different empirical objectives. The OIS estimate is the least-squares solution to an empirical mean-squared objective where the samples are importance weighted:

$$\tilde{v}_g = \arg\min_v \frac{1}{n} \sum_{k=1}^{n} \left( \rho_k Y_k - v \right)^2 \implies \sum_{k=1}^{n} \left( \rho_k Y_k - \tilde{v}_g \right) = 0 \implies \tilde{v}_g = \frac{\sum_{k=1}^{n} \rho_k Y_k}{n} \, . \quad (2)$$

Similarly, the WIS estimate is the least-squares solution to an empirical mean-squared objective where the individual errors are importance weighted:

$$\hat{v}_g = \arg\min_v \frac{1}{n} \sum_{k=1}^{n} \rho_k \left( Y_k - v \right)^2 \implies \sum_{k=1}^{n} \rho_k \left( Y_k - \hat{v}_g \right) = 0 \implies \hat{v}_g = \frac{\sum_{k=1}^{n} \rho_k Y_k}{\sum_{k=1}^{n} \rho_k} \, . \quad (3)$$

We solve similar empirical objectives in a general supervised learning setting with linear function approximation to derive the two new methods.

Consider two correlated random variables $X_k$ and $Y_k$, where $X_k$ takes values from a finite set $\mathcal{X}$, and where $Y_k \in \mathbb{R}$. We want to estimate the conditional expectation of $Y_k$ for each $x \in \mathcal{X}$ under a target distribution $g_{Y|X}$. However, the samples $(X_k, Y_k)$ are generated i.i.d. according to a joint sample distribution $l_{XY}(\cdot)$ with conditional probabilities $l_{Y|X}$ that may differ from the conditional target distribution. Each input is mapped to a feature vector $\phi_k \doteq \phi(X_k) \in \mathbb{R}^m$, and the goal is to estimate the expectation $\mathbb{E}_{g_{Y|X}}[Y_k \mid X_k = x]$ as a linear function of the features

$$\boldsymbol{\theta}^\top \phi(x) \approx v_g(x) \doteq \mathbb{E}_{g_{Y|X}} \left[ Y_k | X_k = x \right] \, .$$

Estimating this expectation is again difficult because the target joint distribution of the input-output pairs $g_{XY}$ can be different than the sample joint distribution $l_{XY}$. Generally, the discrepancy in

the joint distribution may arise from two sources: difference in marginal distribution of inputs, $g_X \neq l_X$, and difference in the conditional distribution of outputs, $g_{Y|X} \neq l_{Y|X}$. Problems where only the former discrepancy arise are known as *covariate shift problems* (Shimodaira 2000). In these problems the conditional expectation of the outputs is assumed unchanged between the target and the sample distributions. In off-policy learning problems, the discrepancy between conditional probabilities is more important. Most off-policy learning methods correct only the discrepancy between the target and the sample conditional distributions of outputs (Hachiya et al. 2009, Maei & Sutton 2010, Yu 2010, Maei 2011, Geist & Scherrer 2014, Dann et al. 2014). In this paper, we also focus only on correcting the discrepancy between the conditional distributions.

The problem of estimating $v_g(x)$ as a linear function of features using samples generated from $l$ can be formulated as the minimization of the mean squared error (MSE) where the solution is as follows:

$$\boldsymbol{\theta}_* \doteq \arg\min_{\boldsymbol{\theta}} \mathbb{E}_{l_X} \left[ \left( \mathbb{E}_{g_{Y|X}} [Y_k|X_k] - \boldsymbol{\theta}^\top \boldsymbol{\phi}_k \right)^2 \right] = \mathbb{E}_{l_X} \left[ \boldsymbol{\phi}_k \boldsymbol{\phi}_k^\top \right]^{-1} \mathbb{E}_{l_X} \left[ \mathbb{E}_{g_{Y|X}} [Y_k|X_k] \boldsymbol{\phi}_k \right]. \quad (4)$$

Similar to the empirical mean-squared objectives defined in (2) and (3), two different empirical objectives can be defined to approximate this solution. In one case the importance weighting is applied to the output samples, $Y_k$, and in the other case the importance weighting is applied to the error, $Y_k - \boldsymbol{\theta}^\top \boldsymbol{\phi}_k$,

$$\tilde{J}_n(\boldsymbol{\theta}) \doteq \frac{1}{n} \sum_{k=1}^n \left( \rho_k Y_k - \boldsymbol{\theta}^\top \boldsymbol{\phi}_k \right)^2 ; \qquad \hat{J}_n(\boldsymbol{\theta}) \doteq \frac{1}{n} \sum_{k=1}^n \rho_k \left( Y_k - \boldsymbol{\theta}^\top \boldsymbol{\phi}_k \right)^2 ,$$

where importance-sampling ratios are defined by $\rho_k \doteq g_{Y|X}(Y_k|X_k)/l_{Y|X}(Y_k|X_k)$.

We can minimize these objectives by equating the derivatives of the above empirical objectives to zero. Provided the relevant matrix inverses exist, the resulting solutions are, respectively

$$\tilde{\boldsymbol{\theta}}_n \doteq \left( \sum_{k=1}^n \boldsymbol{\phi}_k \boldsymbol{\phi}_k^\top \right)^{-1} \sum_{k=1}^n \rho_k Y_k \boldsymbol{\phi}_k \text{, and} \quad (5)$$

$$\hat{\boldsymbol{\theta}}_n \doteq \left( \sum_{k=1}^n \rho_k \boldsymbol{\phi}_k \boldsymbol{\phi}_k^\top \right)^{-1} \sum_{k=1}^n \rho_k Y_k \boldsymbol{\phi}_k . \quad (6)$$

We call $\tilde{\boldsymbol{\theta}}$ the *OIS-LS estimator* and $\hat{\boldsymbol{\theta}}$ the *WIS-LS estimator*.

A least-squares method similar to WIS-LS above was introduced for covariate shift problems by Hachiya, Sugiyama and Ueda (2012). Although superficially similar, that method uses importance-sampling ratios to correct for the discrepancy in the marginal distributions of inputs, whereas WIS-LS corrects for the discrepancy in the conditional expectations of the outputs. For the fully-representable case, unlike WIS-LS, the method of Hachiya et al. becomes an ordinary Monte Carlo estimator with no importance sampling.

## 3 Analysis of the least-squares importance-sampling methods

The two least-squares importance-sampling methods have properties similar to those of the OIS and the WIS methods. In Theorems 1 and 2, we prove that when $v_g$ can be represented as a linear function of the features, then OIS-LS is an unbiased estimator of $\boldsymbol{\theta}_*$, whereas WIS-LS is a biased estimator, similar to the WIS estimator. If the solution is not linearly representable, least-squares methods are not generally unbiased. In Theorem 3 and 4, we show that both least-squares estimators are consistent for $\boldsymbol{\theta}_*$. Finally, we demonstrate that the least-squares methods are generalizations of OIS and WIS by showing, in Theorem 5 and 6, that in the fully representable case (when the features form an orthonormal basis) OIS-LS is equivalent to OIS and WIS-LS is equivalent to WIS.

**Theorem 1.** *If $v_g$ is a linear function of the features, that is, $v_g(x) = \boldsymbol{\theta}_*^\top \boldsymbol{\phi}(x)$, then OIS-LS is an unbiased estimator, that is, $\mathbb{E}_{l_{XY}}[\tilde{\boldsymbol{\theta}}_n] = \boldsymbol{\theta}_*$.*

**Theorem 2.** *Even if $v_g$ is a linear function of the features, that is, $v_g(x) = \boldsymbol{\theta}_*^\top \boldsymbol{\phi}(x)$, WIS-LS is in general a biased estimator, that is, $\mathbb{E}_{l_{XY}}[\hat{\boldsymbol{\theta}}_n] \neq \boldsymbol{\theta}_*$.*

**Theorem 3.** *The OIS-LS estimator $\tilde{\boldsymbol{\theta}}_n$ is a consistent estimator of the MSE solution $\boldsymbol{\theta}_*$ given in (4).*

**Theorem 4.** *The WIS-LS estimator $\hat{\boldsymbol{\theta}}_n$ is a consistent estimator of the MSE solution $\boldsymbol{\theta}_*$ given in (4).*

**Theorem 5.** *If the features form an orthonormal basis, then the OIS-LS estimate $\tilde{\boldsymbol{\theta}}_n^\top \boldsymbol{\phi}(x)$ of input $x$ is equivalent to the OIS estimate of the outputs corresponding to $x$.*

**Theorem 6.** *If the features form an orthonormal basis, then the WIS-LS estimate $\hat{\boldsymbol{\theta}}_n^\top \boldsymbol{\phi}(x)$ of input $x$ is equivalent to the WIS estimate of the outputs corresponding to $x$.*

Proofs of Theorem 1-6 are given in the Appendix.

The WIS-LS estimate is perhaps the most interesting of the two least-squares estimates, because it generalizes WIS to parametric function approximation for the first time and extends its advantages.

## 4 A new off-policy LSTD($\lambda$) with WIS

In sequential decision problems, off-policy learning methods based on important sampling can suffer from the same high-variance issues as discussed above for the supervised case. To address this, we extend the idea of WIS-LS to off-policy reinforcement learning and construct a new off-policy WIS-LSTD($\lambda$) algorithm.

We first explain the problem setting. Consider a learning agent that interacts with an environment where at each step $t$ the state of the environment is $S_t$ and the agent observes a feature vector $\boldsymbol{\phi}_t \doteq \boldsymbol{\phi}(S_t) \in \mathbb{R}^m$. The agent takes an action $A_t$ based on a behavior policy $b(\cdot|S_t)$, that is typically a function of the state features. The environment provides the agent a scalar (reward) signal $R_{t+1}$ and transitions to state $S_{t+1}$. This process continues, generating a trajectory of states, actions and rewards. The goal is to estimate the values of the states under the target policy $\pi$, defined as the expected returns given by the sum of future discounted rewards:

$$v_\pi(s) \doteq \mathbb{E}\left[\sum_{t=0}^\infty R_{t+1} \prod_{k=1}^t \gamma(S_k) \mid S_0 = s, A_t \sim \pi(\cdot|S_t), \forall t\right],$$

where $\gamma(S_k) \in [0, 1]$ is a state-dependent degree of discounting on arrival in $S_k$ (as in Sutton et al. 2014). We assume the rewards and discounting are chosen such that $v_\pi(s)$ is well-defined and finite.

Our primary objective is to estimate $v_\pi$ as a linear function of the features: $v_\pi(s) \approx \boldsymbol{\theta}^\top \boldsymbol{\phi}(s)$, where $\boldsymbol{\theta} \in \mathbb{R}^m$ is a parameter vector to be estimated. As before, we need to correct for the difference in sample distribution resulting from the behavior policy and the target distribution as induced by the target policy. Consider a partial trajectory from time step $k$ to time $t$, consisting of a sequence $S_k, A_k, R_k, S_{k+1}, \ldots, S_t$. The probability of this trajectory occurring given it starts at $S_k$ under the target policy will generally differ from its probability under the behavior policy. The importance-sampling ratio $\rho_k^t$ is defined to be the ratio of these probabilities. This importance-sampling ratio can be written in terms of the product of action-selection probabilities without needing a model of the environment (Sutton & Barto 1998):

$$\rho_k^t \doteq \frac{\prod_{i=k}^{t-1} \pi(A_i|S_i)}{\prod_{i=k}^{t-1} b(A_i|S_i)} = \prod_{i=k}^{t-1} \frac{\pi(A_i|S_i)}{b(A_i|S_i)} = \prod_{i=k}^{t-1} \rho_i,$$

where we use the shorthand $\rho_i \doteq \rho_i^{i+1} = \pi(A_i|S_i)/b(A_i|S_i)$.

We incorporate a common technique to reinforcement learning (RL) where updates are estimated by bootstrapping, fully or partially, on previously constructed state-value estimates. Bootstrapping potentially reduces the variance of the updates compared to using full returns and makes RL algorithms applicable to non-episodic tasks. In this paper we assume that the bootstrapping parameter $\lambda(s) \in [0, 1]$ may depend on the state $s$ (as in Sutton & Barto 1998, Maei & Sutton 2010). In the following, we use the notational shorthands $\gamma_k \doteq \gamma(S_k)$ and $\lambda_k \doteq \lambda(S_k)$.

Following Sutton et al. (2014), we construct an empirical loss as a sum of pairs of squared corrected and uncorrected errors, each corresponding to a different number of steps of lookahead, and each weighted as a function of the intervening discounting and bootstrapping. Let $G_k^t \doteq R_{k+1} + \ldots + R_t$ be the undiscounted return truncated after looking ahead $t - k$ steps. Imagine constructing the

empirical loss for time 0. After leaving $S_0$ and observing $R_1$ and $S_1$, the first uncorrected error is $G_0^1 - \boldsymbol{\theta}^\top \boldsymbol{\phi}_0$, with weight equal to the probability of terminating $1 - \gamma_1$. If we do not terminate, then we continue to $S_1$ and form the first corrected error $G_0^1 + \boldsymbol{v}^\top \boldsymbol{\phi}_1 - \boldsymbol{\theta}^\top \boldsymbol{\phi}_0$ using the bootstrapping estimate $\boldsymbol{v}^\top \boldsymbol{\phi}_1$. The weight on this error is $\gamma_1(1 - \lambda_1)$, and the parameter vector $\boldsymbol{v}$ may differ from $\boldsymbol{\theta}$. Continuing to the next time step, we obtain the second uncorrected error $G_0^2 - \boldsymbol{\theta}^\top \boldsymbol{\phi}_0$ and the second corrected error $G_0^2 + \boldsymbol{v}^\top \boldsymbol{\phi}_2 - \boldsymbol{\theta}^\top \boldsymbol{\phi}_0$, with respective weights $\gamma_1 \lambda_1 (1 - \gamma_2)$ and $\gamma_1 \lambda_1 \gamma_2 (1 - \lambda_2)$. This goes on until we reach the horizon of our data, say at time $t$, when we bootstrap fully with $\boldsymbol{v}^\top \boldsymbol{\phi}_t$, generating a final corrected return error of $G_0^t + \boldsymbol{v}^\top \boldsymbol{\phi}_t - \boldsymbol{\theta}^\top \boldsymbol{\phi}_0$ with weight $\gamma_1 \lambda_1 \cdots \gamma_{t-1} \lambda_{t-1} \gamma_t$. The general form for the uncorrected error is $\epsilon_k^t(\boldsymbol{\theta}) \doteq G_k^t - \boldsymbol{\theta}^\top \boldsymbol{\phi}_k$, and the general form for the corrected error is $\bar{\delta}_k^t(\boldsymbol{\theta}, \boldsymbol{v}) \doteq G_k^t + \boldsymbol{v}^\top \boldsymbol{\phi}_t - \boldsymbol{\theta}^\top \boldsymbol{\phi}_k$. All these errors could be squared, weighted by their weights, and summed to form the overall empirical loss. In the off-policy case, we need to also weight the squares of the errors $\epsilon_k^t$ and $\bar{\delta}_k^t$ by the importance-sampling ratio $\rho_k^t$. Hence, the overall empirical loss at time $k$ for data up to time $t$ can be written as

$$
\ell_k^t(\boldsymbol{\theta}, \boldsymbol{v}) \doteq \rho_k \sum_{i=k+1}^{t-1} C_k^{i-1} \left[ (1 - \gamma_i) \left( \epsilon_k^i(\boldsymbol{\theta}) \right)^2 + \gamma_i (1 - \lambda_i) \left( \bar{\delta}_k^i(\boldsymbol{\theta}, \boldsymbol{v}) \right)^2 \right] \quad , \text{where } C_k^t \doteq \prod_{j=k+1}^{t} \gamma_j \lambda_j \rho_j.
$$
$$
+ \rho_k C_k^{t-1} \left[ (1 - \gamma_t) \left( \epsilon_k^t(\boldsymbol{\theta}) \right)^2 + \gamma_t \left( \bar{\delta}_k^t(\boldsymbol{\theta}, \boldsymbol{v}) \right)^2 \right]
$$

This loss differs from that used by other LSTD($\lambda$) methods in that importance weighting is applied to the individual errors within $\ell_k^t(\boldsymbol{\theta}, \boldsymbol{v})$.

Now, we are ready to state the least-squares problem. As noted by Geist & Scherrer (2014), LSTD($\lambda$) methods can be derived by solving least-squares problems where estimates at each step are matched with multi-step returns starting from those steps given that bootstrapping is done using the solution itself. Our proposed new method, called *WIS-LSTD($\lambda$)*, computes at each time $t$ the solution to the least-squares problem:

$$
\boldsymbol{\theta}_t \doteq \arg \min_{\boldsymbol{\theta}} \sum_{k=0}^{t-1} \ell_k^t(\boldsymbol{\theta}, \boldsymbol{\theta}_t).
$$

At the solution, the derivative of the objective is zero: $\frac{\partial}{\partial \boldsymbol{\theta}} \sum_{k=0}^{t-1} \ell_k^t(\boldsymbol{\theta}, \boldsymbol{\theta}_t) \big|_{\boldsymbol{\theta}=\boldsymbol{\theta}_t} = -\sum_{k=0}^{t-1} 2\delta_{k,t}^\rho(\boldsymbol{\theta}_t, \boldsymbol{\theta}_t) \boldsymbol{\phi}_k = \mathbf{0}$, where the errors $\delta_{k,t}^\rho$ are defined by

$$
\delta_{k,t}^\rho(\boldsymbol{\theta}, \boldsymbol{v}) \doteq \rho_k \sum_{i=k+1}^{t-1} C_k^{i-1} \left[ (1 - \gamma_i) \epsilon_k^i(\boldsymbol{\theta}) + \gamma_i (1 - \lambda_i) \bar{\delta}_k^i(\boldsymbol{\theta}, \boldsymbol{v}) \right]
$$
$$
+ \rho_k C_k^{t-1} \left[ (1 - \gamma_t) \epsilon_k^t(\boldsymbol{\theta}) + \gamma_t \bar{\delta}_k^t(\boldsymbol{\theta}, \boldsymbol{v}) \right].
$$

Next, we separate the terms of $\delta_{k,t}^\rho(\boldsymbol{\theta}_t, \boldsymbol{\theta}_t) \boldsymbol{\phi}_k$ that involve $\boldsymbol{\theta}_t$ from those that do not:

$\delta_{k,t}^\rho(\boldsymbol{\theta}_t, \boldsymbol{\theta}_t) \boldsymbol{\phi}_k = \boldsymbol{b}_{k,t} - \mathbf{A}_{k,t} \boldsymbol{\theta}_t$, where $\boldsymbol{b}_{k,t} \in \mathbb{R}^m$, $\mathbf{A}_{k,t} \in \mathbb{R}^{m \times m}$ and they are defined as

$$
\boldsymbol{b}_{k,t} \doteq \rho_k \sum_{i=k+1}^{t-1} C_k^{i-1} (1 - \gamma_i \lambda_i) G_k^i \boldsymbol{\phi}_k \; + \; \rho_k C_k^{t-1} G_k^t \boldsymbol{\phi}_k,
$$

$$
\mathbf{A}_{k,t} \doteq \rho_k \sum_{i=k+1}^{t-1} C_k^{i-1} \boldsymbol{\phi}_k ((1 - \gamma_i \lambda_i) \boldsymbol{\phi}_k - \gamma_i (1 - \lambda_i) \boldsymbol{\phi}_i)^\top \; + \; \rho_k C_k^{t-1} \boldsymbol{\phi}_k (\boldsymbol{\phi}_k - \gamma_t \boldsymbol{\phi}_t)^\top.
$$

Therefore, the solution can be found as follows:

$$
\sum_{k=0}^{t-1} (\boldsymbol{b}_{k,t} - \mathbf{A}_{k,t} \boldsymbol{\theta}_t) = \mathbf{0} \implies \boldsymbol{\theta}_t = \mathbf{A}_t^{-1} \boldsymbol{b}_t, \text{ where } \mathbf{A}_t \doteq \sum_{k=0}^{t-1} \mathbf{A}_{k,t}, \quad \boldsymbol{b}_t \doteq \sum_{k=0}^{t-1} \boldsymbol{b}_{k,t}. \tag{7}
$$

In the following we show that WIS-LS is a special case of the above algorithm defined by (7). As Theorem 6 shows that WIS-LS generalizes WIS, it follows that the above algorithm generalizes WIS as well.

**Theorem 7.** *At termination, the algorithm defined by* (7) *is equivalent to the WIS-LS method in the sense that if $\lambda_0 = \cdots = \lambda_t = \gamma_0 = \cdots = \gamma_{t-1} = 1$ and $\gamma_t = 0$, then $\boldsymbol{\theta}_t$ defined in (7) equals $\hat{\boldsymbol{\theta}}_t$ as defined in (6), with $Y_k \doteq G_k^t$.* (Proved in the Appendix).

Our last challenge is to find an equivalent efficient online algorithm for this method. The solution in (7) cannot be computed incrementally in this form. When a new sample arrives at time $t+1$, $\mathbf{A}_{k,t+1}$ and $\boldsymbol{b}_{k,t+1}$ have to be computed for each $k = 0, \ldots, t$, and hence the computational complexity of this solution grows with time. It would be preferable if the solution at time $t+1$ could be computed incrementally based on the estimates from time $t$, requiring only constant computational complexity per time step. It is not immediately obvious such an efficient update exists. For instance, for $\lambda = 1$ this method achieves full Monte Carlo (weighted) importance-sampling estimation, which means whenever the target policy deviates from the behavior policy all previously made updates have to be unmade so that no updates are made towards a trajectory which is impossible under the target policy. Sutton et al. (2014) show it is possible to derive efficient updates in some cases with the use of *provisional* parameters which keep track of the provisional updates that might need to be unmade when a deviation occurs. In the following, we show that using such provisional parameters it is also possible to achieve an equivalent efficient update for (7).

We first write both $\boldsymbol{b}_{k,t}$ and $\mathbf{A}_{k,t}$ recursively in $t$ (derivations in Appendix A.8):

$$\boldsymbol{b}_{k,t+1} = \boldsymbol{b}_{k,t} + \rho_k C_k^t R_{t+1} \boldsymbol{\phi}_k + (\rho_t - 1)\gamma_t \lambda_t \rho_k C_k^{t-1} G_k^t \boldsymbol{\phi}_k,$$

$$\mathbf{A}_{k,t+1} = \mathbf{A}_{k,t} + \rho_k C_k^t \boldsymbol{\phi}_k (\boldsymbol{\phi}_t - \gamma_{t+1}\boldsymbol{\phi}_{t+1})^\top + (\rho_t - 1)\gamma_t \lambda_t \rho_k C_k^{t-1} \boldsymbol{\phi}_k(\boldsymbol{\phi}_k - \boldsymbol{\phi}_t)^\top.$$

Using the above recursions, we can write the updates of both $\boldsymbol{b}_t$ and $\mathbf{A}_t$ incrementally. The vector $\boldsymbol{b}_t$ can be updated incrementally as

$$\boldsymbol{b}_{t+1} = \sum_{k=0}^{t} \boldsymbol{b}_{k,t+1} = \sum_{k=0}^{t-1} \boldsymbol{b}_{k,t+1} + \boldsymbol{b}_{t,t+1} = \sum_{k=1}^{t-1} \boldsymbol{b}_{k,t} + \rho_t R_{t+1} \boldsymbol{\phi}_t + R_{t+1} \sum_{k=1}^{t-1} \rho_k C_k^t \boldsymbol{\phi}_k$$

$$+ (\rho_t - 1)\gamma_t \lambda_t \sum_{k=1}^{t-1} \rho_k C_k^{t-1} G_k^t \boldsymbol{\phi}_k \qquad\qquad = \boldsymbol{b}_t + R_{t+1}\boldsymbol{e}_t + (\rho_t - 1)\boldsymbol{u}_t, \qquad (8)$$

where the eligibility trace $\boldsymbol{e}_t \in \mathbb{R}^m$ and the provisional vector $\boldsymbol{u}_t \in \mathbb{R}^m$ are defined as follows:

$$\boldsymbol{e}_t = \rho_t \boldsymbol{\phi}_t + \sum_{k=1}^{t-1} \rho_k C_k^t \boldsymbol{\phi}_k = \rho_t \boldsymbol{\phi}_t + \rho_t \gamma_t \lambda_t \left( \rho_{t-1}\boldsymbol{\phi}_{t-1} + \sum_{k=1}^{t-2} \rho_k C_k^{t-1} \boldsymbol{\phi}_k \right) = \rho_t(\boldsymbol{\phi}_t + \gamma_t \lambda_t \boldsymbol{e}_{t-1}), \quad (9)$$

$$\boldsymbol{u}_t = \gamma_t \lambda_t \sum_{k=1}^{t-1} \rho_k C_k^{t-1} G_k^t \boldsymbol{\phi}_k = \gamma_t \lambda_t \left( \rho_{t-1}\gamma_{t-1}\lambda_{t-1} \sum_{k=1}^{t-1} \rho_k C_k^{t-2} G_k^{t-1} \boldsymbol{\phi}_k \right.$$

$$\left. + R_t \sum_{k=1}^{t-2} \rho_k C_k^{t-1} \boldsymbol{\phi}_k + \rho_{t-1} R_t \boldsymbol{\phi}_{t-1} \right) \qquad = \gamma_t \lambda_t \left( \rho_{t-1}\boldsymbol{u}_{t-1} + R_t \boldsymbol{e}_{t-1} \right). \qquad (10)$$

The matrix $\mathbf{A}_t$ can be updated incrementally as

$$\mathbf{A}_{t+1} = \sum_{k=0}^{t} \mathbf{A}_{k,t+1} = \sum_{k=0}^{t-1} \mathbf{A}_{k,t+1} + \mathbf{A}_{t,t+1} = \sum_{k=0}^{t-1} \mathbf{A}_{k,t} + \rho_t \boldsymbol{\phi}_t(\boldsymbol{\phi}_t - \gamma_{t+1}\boldsymbol{\phi}_{t+1})^\top$$

$$+ \sum_{k=1}^{t-1} \rho_k C_k^t \boldsymbol{\phi}_k(\boldsymbol{\phi}_t - \gamma_{t+1}\boldsymbol{\phi}_{t+1})^\top + (\rho_t - 1)\gamma_t \lambda_t \sum_{k=1}^{t-1} \rho_k C_k^{t-1} \boldsymbol{\phi}_k(\boldsymbol{\phi}_k - \boldsymbol{\phi}_t)^\top$$

$$= \mathbf{A}_t + \boldsymbol{e}_t(\boldsymbol{\phi}_t - \gamma_{t+1}\boldsymbol{\phi}_{t+1})^\top + (\rho_t - 1)\mathbf{V}_t, \qquad (11)$$

where the provisional matrix $\mathbf{V}_t \in \mathbb{R}^{m \times m}$ is defined as

$$\mathbf{V}_t = \gamma_t \lambda_t \sum_{k=1}^{t-1} \rho_k C_k^{t-1} \boldsymbol{\phi}_k(\boldsymbol{\phi}_k - \boldsymbol{\phi}_t)^\top = \gamma_t \lambda_t \left( \rho_{t-1}\gamma_{t-1}\lambda_{t-1} \sum_{k=1}^{t-2} \rho_k C_k^{t-2} \boldsymbol{\phi}_k(\boldsymbol{\phi}_k - \boldsymbol{\phi}_{t-1})^\top \right.$$

$$\left. + \sum_{k=1}^{t-2} \rho_k C_k^{t-1} \boldsymbol{\phi}_k(\boldsymbol{\phi}_{t-1} - \boldsymbol{\phi}_t)^\top + \rho_{t-1}\boldsymbol{\phi}_{t-1}(\boldsymbol{\phi}_{t-1} - \boldsymbol{\phi}_t)^\top \right)$$

$$= \gamma_t \lambda_t \left( \rho_{t-1}\mathbf{V}_{t-1} + \boldsymbol{e}_{t-1}(\boldsymbol{\phi}_{t-1} - \boldsymbol{\phi}_t)^\top \right). \qquad (12)$$

Then the parameter vector can be updated as: $\boldsymbol{\theta}_{t+1} = (\mathbf{A}_{t+1})^{-1} \boldsymbol{b}_{t+1}.$ \qquad (13)

Equations (8–13) comprise our WIS-LSTD($\lambda$). Its per-step computational complexity is $O(m^3)$, where $m$ is the number of features. The computational cost of this method does not increase with time. At present we are unsure whether or not there is an $O(m^2)$ implementation.

**Theorem 8.** *The off-policy LSTD($\lambda$) method defined in (8–13) is equivalent to the off-policy LSTD($\lambda$) method defined in (7) in the sense that they compute the same $\boldsymbol{\theta}_t$ at each time $t$.*

*Proof.* The result follows immediately from the above derivation.

It is easy to see that in the on-policy case this method becomes equivalent to on-policy LSTD($\lambda$) (Boyan 1999) by noting that the third term of both $\boldsymbol{b}_t$ and $\mathbf{A}_t$ updates in (8) and (11) becomes zero, because in the on-policy case all the importance-sampling ratios are 1.

Recently Dann et al. (2014) proposed another least-squares based off-policy method called recursive LSTD-TO($\lambda$). Unlike our algorithm, that algorithm does not specialize to WIS in the fully representable case, and it does not seem as closely related to WIS. The Adaptive Per-Decision Importance Weighting (APDIW) method by Hachiya et al. (2009) is superficially similar to WIS-LSTD($\lambda$), there are several important differences. APDIW is a one-step method that always fully bootstraps whereas WIS-LSTD($\lambda$) covers the full spectrum of multi-step backups including both one-step backup and Monte Carlo update. In the fully representable case, APDIW does not become equivalent to the WIS estimate, whereas WIS-LSTD(1) does. Moreover, APDIW does not find a consistent estimation of the off-policy target whereas WIS algorithms do.

## 5 Experimental results

We compared the performance of the proposed WIS-LSTD($\lambda$) method with the conventional off-policy LSTD($\lambda$) by Yu (2010) on two random-walk tasks for off-policy policy evaluation. These random-walk tasks consist of a Markov chain with 11 non-terminal and two terminal states. They can be imagined to be laid out horizontally, where the two terminal states are at the left and the right ends of the chain. From each non-terminal state, there are two actions available: *left*, which leads to the state to the left and *right*, which leads to the state to the right. The reward is 0 for all transitions except for the rightmost transition to the terminal state, where it is +1. The initial state was set to the state in the middle of the chain. The behavior policy chooses an action uniformly randomly, whereas the target policy chooses the *right* action with probability 0.99. The termination function $\gamma$ was set to 1 for the non-terminal states and 0 for the terminal states.

We used two tasks based on this Markov chain in our experiments. These tasks differ by how the non-terminal states were mapped to features. The terminal states were always mapped to a vector with all zero elements. For each non-terminal state, the features were normalized so that the $L^2$ norm of each feature vector was one. For the first task, the feature representation was *tabular*, that is, the feature vectors were standard basis vectors. In this representation, each feature corresponded to only one state. For the second task, the feature vectors were binary representations of state indices. There were 11 non-terminal states, hence each feature vector had $\lfloor \log_2(11) \rfloor + 1 = 4$ components. These vectors for the states from left to right were $(0, 0, 0, 1)^\top, (0, 0, 1, 0)^\top, (0, 0, 1, 1)^\top, \ldots, (1, 0, 1, 1)^\top$, which were then normalized to get unit vectors. These features heavily underrepresented the states, due to the fact that 11 states were represented by only 4 features.

We tested both algorithms for different values of constant $\lambda$, from 0 to 0.9 in steps of 0.1 and from 0.9 to 1.0 in steps of 0.025. The matrix to be inverted in both methods was initialized to $\epsilon \mathbf{I}$, where the regularization parameter $\epsilon$ was varied by powers of 10 with powers chosen from -3 to +3 in steps of 0.2. Performance was measured as the empirical mean squared error (MSE) between the estimated value of the initial state and its true value under the target policy projected to the space spanned by the given features. This error was measured at the end of each of 200 episodes for 100 independent runs.

Figure 1 shows the results for the two tasks in terms of empirical convergence rate, optimum performance and parameter sensitivity. Each curve shows MSE together with standard errors. The first row shows results for the tabular task and the second row shows results for the function approximation task. The first column shows learning curves using $(\lambda, \epsilon) = (0, 1)$ for the first task and $(0.95, 10)$ for the second. It shows that in both cases WIS-LSTD($\lambda$) learned faster and gave lower error throughout the period of learning. The second column shows performance with respect to different $\lambda$ optimized over $\epsilon$. The x-axis is plotted in a reverse log scale, where higher values are more spread out than the lower values. In both tasks, WIS-LSTD($\lambda$) outperformed the conventional LSTD($\lambda$) for all values of $\lambda$. For the best parameter setting (best $\lambda$ and $\epsilon$), WIS-LSTD($\lambda$) outperformed LSTD($\lambda$) by an order

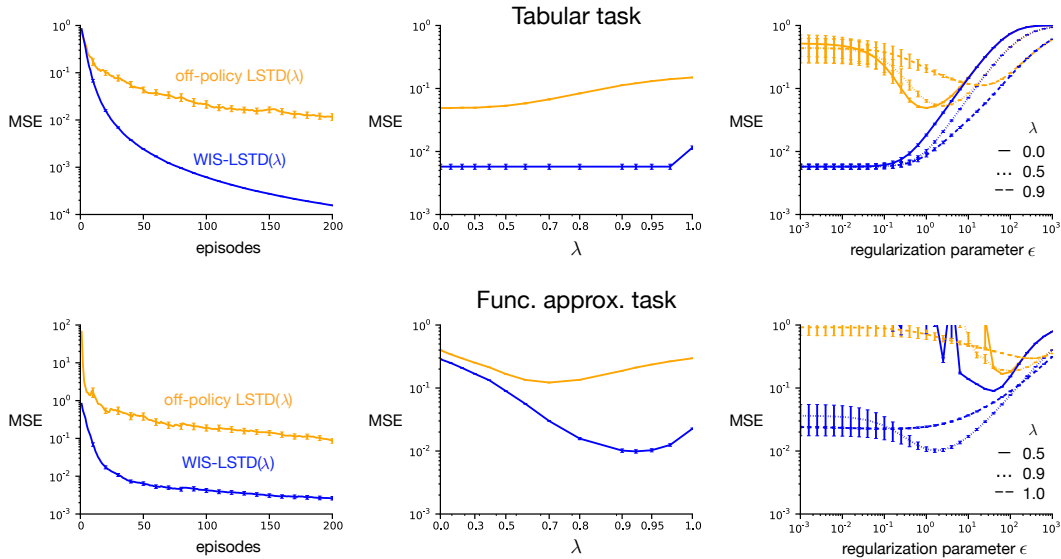

Figure 1: Empirical comparison of our WIS-LSTD($\lambda$) with conventional off-policy LSTD($\lambda$) on two random-walk tasks. The empirical Mean Squared Error shown is for the initial state at the end of each episode, averaged over 100 independent runs (and also over 200 episodes in column 2 and 3).

of magnitude. The third column shows performance with respect to the regularization parameter $\epsilon$ for three representative values of $\lambda$. For a wide range of $\epsilon$, WIS-LSTD($\lambda$) outperformed conventional LSTD($\lambda$) by an order of magnitude. Both methods performed similarly for large $\epsilon$, as such large values essentially prevent learning for a long period of time. In the function approximation task when smaller values of $\epsilon$ were chosen, $\lambda$ close to 1 led to more stable estimates, whereas smaller $\lambda$ introduced high variance for both methods. In both tasks, the better-performing regions of $\epsilon$ (the U-shaped depressions) were wider for WIS-LSTD($\lambda$).

# 6   Conclusion

Although importance sampling is essential to off-policy learning and has become a key part of modern reinforcement learning algorithms, its most effective form—WIS—has been neglected because of the difficulty of combining it with parametric function approximation. In this paper, we have begun to overcome these difficulties. First, we have shown that the WIS estimate can be viewed as the solution to an empirical objective where the squared errors of individual samples are weighted by the importance-sampling ratios. Second, we have introduced a new method for general supervised learning called WIS-LS by extending the error-weighted empirical objective to linear function approximation and shown that the new method has similar properties as those of the WIS estimate. Finally, we have introduced a new off-policy LSTD algorithm WIS-LSTD($\lambda$) that extends the benefits of WIS to reinforcement learning. Our empirical results show that the new WIS-LSTD($\lambda$) can outperform Yu's off-policy LSTD($\lambda$) in both tabular and function approximation tasks and shows robustness in terms of its parameters. An interesting direction for future work is to extend these ideas to off-policy linear-complexity methods.

## Acknowledgement

This work was supported by grants from Alberta Innovates Technology Futures, National Science and Engineering Research Council, and Alberta Innovates Centre for Machine Learning.

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
