[Supplementary Material · normalization10-submitted-appendix.pdf]

## Appendix

### A.1 Proof of Theorem 1

**Theorem 1.** *If $v_g$ is a linear function of the features, that is, $v_g(x) = \boldsymbol{\theta}_*^\top \boldsymbol{\phi}(x)$, then OIS-LS is an unbiased estimator, that is, $\mathbb{E}_l[\tilde{\boldsymbol{\theta}}_n] = \boldsymbol{\theta}_*$.*

*Proof.* The proof is given by the following derivation:

$$
\mathbb{E}_l[\tilde{\boldsymbol{\theta}}_n] = \mathbb{E}_l\left[ \left(\sum_{k=1}^n \boldsymbol{\phi}_k \boldsymbol{\phi}_k^\top\right)^{-1} \sum_{k=1}^n \rho_k Y_k \boldsymbol{\phi}_k \right]
$$

$$
= \mathbb{E}_{l_X}\left[ \left(\sum_{k=1}^n \boldsymbol{\phi}_k \boldsymbol{\phi}_k^\top\right)^{-1} \sum_{k=1}^n \mathbb{E}_{l_{Y|X}}\left[\rho_k Y_k | X_k\right] \boldsymbol{\phi}_k \right]
$$

$$
= \mathbb{E}_{l_X}\left[ \left(\sum_{k=1}^n \boldsymbol{\phi}_k \boldsymbol{\phi}_k^\top\right)^{-1} \sum_{k=1}^n \mathbb{E}_{g_{Y|X}}\left[Y_k | X_k\right] \boldsymbol{\phi}_k \right] = \mathbb{E}_{l_X}\left[ \left(\sum_{k=1}^n \boldsymbol{\phi}_k \boldsymbol{\phi}_k^\top\right)^{-1} \sum_{k=1}^n v_g(X_k)\boldsymbol{\phi}_k \right]
$$

$$
= \mathbb{E}_{l_X}\left[ \left(\sum_{k=1}^n \boldsymbol{\phi}_k \boldsymbol{\phi}_k^\top\right)^{-1} \sum_{k=1}^n \boldsymbol{\phi}_k \boldsymbol{\phi}_k^\top \boldsymbol{\theta}_* \right] = \mathbb{E}_{l_X}\left[ \left(\sum_{k=1}^n \boldsymbol{\phi}_k \boldsymbol{\phi}_k^\top\right)^{-1} \left(\sum_{k=1}^n \boldsymbol{\phi}_k \boldsymbol{\phi}_k^\top\right) \right] \boldsymbol{\theta}_* = \boldsymbol{\theta}_*. \quad \square
$$

### A.2 Proof of Theorem 2

**Theorem 2.** *Even if $v_g$ is a linear function of the features, that is, $v_g(x) = \boldsymbol{\theta}_*^\top \boldsymbol{\phi}(x)$, the WIS-LS estimator defined in (6) is a biased estimator, that is, $\mathbb{E}_l[\hat{\boldsymbol{\theta}}_n] \neq \boldsymbol{\theta}_*$.*

*Proof.* : We prove it by providing a counterexample to the claim that $\mathbb{E}_l[\hat{\boldsymbol{\theta}}_n] = \boldsymbol{\theta}_*$. Consider $\mathcal{X} = \{x\}$ and $\phi(x) = 1$. It is easy to see that in this case $v_g = \theta_* = \mathbb{E}_g[Y_k]$. Then the WIS-LS estimator $\hat{\theta}_n$ reduces to the WIS estimator:

$$
\hat{\theta}_n = \left(\sum_{k=1}^n \rho_k\right)^{-1} \sum_{k=1}^n \rho_k Y_k = \hat{v}_g,
$$

which is a biased estimator, that is, $\mathbb{E}_l[\hat{v}_g] \neq v_g$. Hence, in general, $\mathbb{E}_l[\hat{\boldsymbol{\theta}}_n] \neq \boldsymbol{\theta}_*$. $\quad\square$

### A.3 Proof of Theorem 3

**Theorem 3.** *The OIS-LS estimator $\tilde{\boldsymbol{\theta}}_n$ is a consistent estimator of the MSE solution $\boldsymbol{\theta}_*$ given in (4).*

*Proof.* Due to the strong law of large numbers

$$
\frac{1}{n}\sum_{k=1}^n \boldsymbol{\phi}_k \boldsymbol{\phi}_k^\top \xrightarrow{w.p.1} \mathbb{E}_{l_X}\left[\boldsymbol{\phi}_k \boldsymbol{\phi}_k^\top\right]; \quad \frac{1}{n}\sum_{k=1}^n \rho_k Y_k \boldsymbol{\phi}_k \xrightarrow{w.p.1} \mathbb{E}_l\left[\rho_k Y_k \boldsymbol{\phi}_k\right] = \mathbb{E}_{l_X}\left[\mathbb{E}_{g_{Y|X}}\left[Y_k | X_k\right]\boldsymbol{\phi}_k\right].
$$

Then it follows that $\tilde{\boldsymbol{\theta}}_n \xrightarrow{w.p.1} \boldsymbol{\theta}_*$. $\quad\square$

### A.4 Proof of Theorem 4

**Theorem 4.** *The WIS-LS estimator $\hat{\boldsymbol{\theta}}_n$ is a consistent estimator of the MSE solution $\boldsymbol{\theta}_*$ given in (4).*

*Proof.* It is very similar to the above proof. The only difference is that here we have to show $\frac{1}{n}\sum_{k=1}^n \rho_k \boldsymbol{\phi}_k \boldsymbol{\phi}_k^\top \xrightarrow{w.p.1} \mathbb{E}_{l_X}\left[\boldsymbol{\phi}_k \boldsymbol{\phi}_k^\top\right]$. However, it again follows due to the strong law of large numbers noting that $\mathbb{E}_{l_{XY}}\left[\rho_k \boldsymbol{\phi}_k \boldsymbol{\phi}_k^\top\right] = \mathbb{E}_{l_X}\left[\mathbb{E}_{l_{Y|X}}\left[\rho_k | X_k\right]\boldsymbol{\phi}_k \boldsymbol{\phi}_k^\top\right] = \mathbb{E}_{l_X}\left[\boldsymbol{\phi}_k \boldsymbol{\phi}_k^\top\right]$. $\quad\square$

### A.5 Proof of Theorem 5

**Theorem 5.** *If the features form an orthonormal basis, then the OIS-LS estimate $\tilde{\boldsymbol{\theta}}_n^\top \phi(x)$ of input $x$ is equivalent to the OIS estimate of the outputs corresponding to $x$.*

*Proof.* Let $\boldsymbol{\Phi}$ denote to be the feature matrix the rows of which contain the feature vectors of different unique inputs: $\boldsymbol{\Phi} = \left(\phi(x_1), \ldots, \phi(x_{|\mathcal{X}|})\right)^\top$, where $x_1, \ldots, x_{|\mathcal{X}|}$ are different unique inputs. Then the vector containing the estimated conditional expectation of outputs for each unique input according to the OIS-LS estimator can be written as

$$\boldsymbol{\Phi}\tilde{\boldsymbol{\theta}}_n = \boldsymbol{\Phi}\left(\sum_{x\in\mathcal{X}} n_x \phi(x)\phi(x)^\top\right)^{-1} \sum_{x\in\mathcal{X}}\left(\sum_{i=1}^{n_x} \rho_{x,i} Y_{x,i}\right)\phi(x) = \boldsymbol{\Phi}\left(\boldsymbol{\Phi}^\top \mathbf{N}\boldsymbol{\Phi}\right)^{-1}\boldsymbol{\Phi}^\top \boldsymbol{y},$$

where $n_x$ is the number of times input $x$ is observed among $n$ samples, $Y_{x,i}$ is the output corresponding to the $i$th occurrence of input $x$ and $\rho_{x,i}$ is the corresponding importance-sampling ratio. Here, $\mathbf{N}$ is a diagonal matrix where the $i$th diagonal element contains $n_{x_i}$: $\mathbf{N} = \operatorname{diag}\left(n_{x_1}, \ldots, n_{x_{|\mathcal{X}|}}\right)$ and $\boldsymbol{y} = \left(\sum_{i=1}^{n_{x_1}} \rho_{x_1,i} Y_{x_1,i}, \ldots, \sum_{i=1}^{n_{|\mathcal{X}|}} \rho_{|\mathcal{X}|,i} Y_{|\mathcal{X}|,i}\right)^\top$.

Note that, due to orthonormality of the features, $\boldsymbol{\Phi}$ is necessarily a square matrix and full rank. Therefore, it follows that the vector of the estimates can be written as

$$\boldsymbol{\Phi}\tilde{\boldsymbol{\theta}}_n = \boldsymbol{\Phi}\boldsymbol{\Phi}^{-1}\mathbf{N}^{-1}\boldsymbol{\Phi}^{\top -1}\boldsymbol{\Phi}^\top \boldsymbol{y} = \mathbf{N}^{-1}\boldsymbol{y}.$$

The element of this vector corresponding to any input $x$ is the ordinary importance-sampling estimator of its corresponding outputs: $n_x^{-1}\sum_{i=1}^{n_x}\rho_{x,i}Y_{x,i}$. $\qquad\square$

### A.6 Proof of Theorem 6

**Theorem 6.** *If the features form an orthonormal basis, then the WIS-LS estimate $\hat{\boldsymbol{\theta}}_n^\top \phi(x)$ of input $x$ is equivalent to the WIS estimate of the outputs corresponding to $x$.*

*Proof.* The proof is similar to the proof of Theorem 5. First, we write the vector of the estimates according to the WIS-LS estimate as

$$\boldsymbol{\Phi}\hat{\boldsymbol{\theta}}_n = \boldsymbol{\Phi}\left(\sum_{x\in\mathcal{X}}\left(\sum_{i=1}^{n_x}\rho_{x,i}\right)\phi(x)\phi(x)^\top\right)^{-1}\sum_{x\in\mathcal{X}}\left(\sum_{i=1}^{n_x}\rho_{x,i}Y_{x,i}\right)\phi(x) = \boldsymbol{\Phi}\left(\boldsymbol{\Phi}^\top \mathbf{R}\boldsymbol{\Phi}\right)^{-1}\boldsymbol{\Phi}^\top \boldsymbol{y},$$

where $\mathbf{R}$ is a diagonal matrix with each diagonal element containing the total summation of the importance-sampling ratios corresponding to each input: $\mathbf{R} = \operatorname{diag}\left(\left(\sum_{i=1}^{n_{x_1}}\rho_{x_1,i}\right), \ldots, \left(\sum_{i=1}^{n_{x_{|\mathcal{X}|}}}\rho_{x_{|\mathcal{X}|},i}\right)\right)$. Hence, the vector of estimates can be written as

$$\boldsymbol{\Phi}\hat{\boldsymbol{\theta}}_n = \boldsymbol{\Phi}\boldsymbol{\Phi}^{-1}\mathbf{R}^{-1}\boldsymbol{\Phi}^{\top -1}\boldsymbol{\Phi}^\top \boldsymbol{y} = \mathbf{R}^{-1}\boldsymbol{y},$$

The element of this vector corresponding to any input $x$ is the WIS estimate of its corresponding outputs: $\left(\sum_{i=1}^{n_x}\rho_{x,i}\right)^{-1}\sum_{i=1}^{n_x}\rho_{x,i}Y_{x,i}$. $\qquad\square$

### A.7 Proof of Theorem 7

**Theorem 7.** *At termination, the algorithm defined by (7) is equivalent to the WIS-LS method in the sense that if $\lambda_0 = \cdots = \lambda_t = \gamma_0 = \cdots = \gamma_{t-1} = 1$ and $\gamma_t = 0$, then $\boldsymbol{\theta}_t$ defined in (7) equals $\hat{\boldsymbol{\theta}}_t$ as defined in (6), with $Y_k \doteq G_k^t$.*

*Proof.* When $\gamma_0 = \cdots = \gamma_{t-1} = 1$, $\gamma_t = 0$ and also $\lambda_0 = \cdots = \lambda_t = 1$, then

$$\boldsymbol{b}_{k,t} = \prod_{j=k}^{t-1}\rho_j G_k^t \phi_k = \rho_k^t G_k^t \phi_k, \qquad \mathbf{A}_{k,t} = \prod_{j=k}^{t-1}\rho_j \phi_k \phi_k^\top = \rho_k^t \phi_k \phi_k^\top.$$

Hence, the solution can be written as $\boldsymbol{\theta}_t = \mathbf{A}_t^{-1}\boldsymbol{b}_t = \left(\sum_{k=0}^{t-1}\mathbf{A}_{k,t}\right)^{-1}\sum_{k=0}^{t-1}\boldsymbol{b}_{k,t} = \left(\sum_{k=0}^{t-1}\rho_k^t \phi_k \phi_k^\top\right)^{-1}\sum_{k=0}^{t-1}\rho_k^t G_k^t \phi_k$, which is the WIS-LS solution. $\qquad\square$

## A.8 Derivations of the recursive updates of $b_{k,t}$ and $\mathbf{A}_{k,t}$ in $t$

The derivations are given below:

$$
\begin{aligned}
\boldsymbol{b}_{k,t+1} &= \rho_k \sum_{i=k+1}^{t} C_k^{i-1}(1-\gamma_i\lambda_i)G_k^i\boldsymbol{\phi}_k + \rho_k C_k^t G_k^{t+1}\boldsymbol{\phi}_k \\
&= \rho_k \sum_{i=k+1}^{t-1} C_k^{i-1}(1-\gamma_i\lambda_i)G_k^i\boldsymbol{\phi}_k + (1-\gamma_t\lambda_t)\rho_k C_k^{t-1}G_k^t\boldsymbol{\phi}_k \\
&\quad + \rho_t\gamma_t\lambda_t\rho_k C_k^{t-1}\left(G_k^t + R_{t+1}\right)\boldsymbol{\phi}_k \\
&= \boldsymbol{b}_{k,t} + \rho_k C_k^t R_{t+1}\boldsymbol{\phi}_k + (\rho_t - 1)\gamma_t\lambda_t\rho_k C_k^{t-1}G_k^t\boldsymbol{\phi}_k, \\[6pt]
\mathbf{A}_{k,t+1} &= \rho_k \sum_{i=k+1}^{t} C_k^{i-1}\boldsymbol{\phi}_k((1-\gamma_i\lambda_i)\boldsymbol{\phi}_k - \gamma_i(1-\lambda_i)\boldsymbol{\phi}_i)^\top + \rho_k C_k^t\boldsymbol{\phi}_k(\boldsymbol{\phi}_k - \gamma_{t+1}\boldsymbol{\phi}_{t+1})^\top \\
&= \rho_k \sum_{i=k+1}^{t-1} C_k^{i-1}\boldsymbol{\phi}_k((1-\gamma_i\lambda_i)\boldsymbol{\phi}_k - \gamma_i(1-\lambda_i)\boldsymbol{\phi}_i)^\top \\
&\quad + \rho_k C_k^{t-1}\boldsymbol{\phi}_k((1-\gamma_t\lambda_t)\boldsymbol{\phi}_k - \gamma_t(1-\lambda_t)\boldsymbol{\phi}_t)^\top + \rho_k C_k^t\boldsymbol{\phi}_k(\boldsymbol{\phi}_k - \gamma_{t+1}\boldsymbol{\phi}_{t+1})^\top \\
&= \rho_k \sum_{i=k+1}^{t-1} C_k^{i-1}\boldsymbol{\phi}_k((1-\gamma_i\lambda_i)\boldsymbol{\phi}_k - \gamma_i(1-\lambda_i)\boldsymbol{\phi}_i)^\top + \rho_k C_k^{t-1}\boldsymbol{\phi}_k(\boldsymbol{\phi}_k - \gamma_t\boldsymbol{\phi}_t)^\top \\
&\quad + \rho_k C_k^t\boldsymbol{\phi}_k(\boldsymbol{\phi}_k - \gamma_{t+1}\boldsymbol{\phi}_{t+1})^\top - \gamma_t\lambda_t\rho_k C_k^{t-1}\boldsymbol{\phi}_k(\boldsymbol{\phi}_k - \boldsymbol{\phi}_t)^\top \\
&= \mathbf{A}_{k,t} + \rho_k C_k^t\boldsymbol{\phi}_k(\boldsymbol{\phi}_k - \boldsymbol{\phi}_t + \boldsymbol{\phi}_t - \gamma_{t+1}\boldsymbol{\phi}_{t+1})^\top - \gamma_t\lambda_t\rho_k C_k^{t-1}\boldsymbol{\phi}_k(\boldsymbol{\phi}_k - \boldsymbol{\phi}_t)^\top \\
&= \mathbf{A}_{k,t} + \rho_k C_k^t\boldsymbol{\phi}_k(\boldsymbol{\phi}_t - \gamma_{t+1}\boldsymbol{\phi}_{t+1})^\top + (\rho_t - 1)\gamma_t\lambda_t\rho_k C_k^{t-1}\boldsymbol{\phi}_k(\boldsymbol{\phi}_k - \boldsymbol{\phi}_t)^\top.
\end{aligned}
$$