[Reviews · NeurIPS 2014]

Submitted by Assigned_Reviewer_4

This is a very well-written paper that explores the use of weighted importance sampling to speed up learning in off-policy LSTD-type algorithms. The theoretical results are solid and what one would expect. The computational results are striking. The technique could serve as a useful component in design of RL algorithms.
Summary: The paper is very well-written and presents a useful idea validated by striking computational results.

Submitted by Assigned_Reviewer_12

This paper provides an application of important sampling with heuristic variance reduction technique, which has been already applied to TD learning, into value function approximation. The unbiased condition in value function approximation is theoretically discussed. The performance of the proposed method was evaluated using simple Markov chains.

Quality, clarity, originality and significance:
This paper is written well but the motivation of this paper is not so clear compared to previous works. That is, the same variance reduction technique has been already applied to a TD based value function learning by Precup 2000 and the importance weighting in value function approximation with variance reduction has been already explored by Hachiya 2009:
Adaptive Importance Sampling for Value Function Approximation in Off-policy Reinforcement Learning
Hachiya et al., Neural Networks, vol.22, no.10, pp.1399–1410, 2009.

Although Theorems for the unbiased estimator in linear model would be original and little useful for readers, I cannot find any significant contribution in this paper compared to previous work. For the acceptance, I think the author needs to introduce new type of variance reduction method in value function approximation and well explain its value and compare its performance with previous works such as Precup 2000 and Hachiya 2009, etc.
Summary: This paper provides an application of important sampling with heuristic variance reduction technique, which has been already applied to TD learning, into value function approximation. The unbiased condition in value function approximation is theoretically discussed. The performance of the proposed method was evaluated using simple Markov chains.

Submitted by Assigned_Reviewer_42

Summary
=============

The paper brings the idea of weighted importance sampling (WIS) to off-policy policy evaluation. The core of the work is built around the observation that WIS can be formulated as a weighted regression problem and the corresponding formulation is extended to the problem of linear approximation in off-policy policy evaluation. The soundness of the method is discussed in detail and its empirical effectiveness is evaluated in a simple chain problem.

Quality
==============

The way the paper is written may give the impression that the application of WIS is very simple and almost obvious. Nonetheless, as remarked by the authors, bringing WIS to off-policy evaluation was not trivial and it required a deep understanding of the structure of WIS.

The main contributions of the paper are:
1- I found the formulation of OIS and WIS in eq. 2 and 3 and their extension to linear regression very simple but elegant and very insightful. Eqs. 5 and 6 clearly reveal how the idea of OIS and WIS estimators smoothly extend to the case of linear regression. This is further supported by the analysis in Section 3. There are only two aspects that I would like the authors to clarify further:
a- The definition of the OIS and WIS linear estimators only correct for the unbalance in the output distribution, while the possible difference in the input distributions is not taken into account (starting from the definition of J^{MSE}). Why? How would you correct differences in the input distribution?
b- In the analysis in Section 3 no discussion is provided about whether WIS-LS may actually benefit from similar variance-reduction properties as for WIS. It would be helpful to report in sections 1/2 a more clear overview of the theoretical results available for WIS and discuss in section 3 how much these results may extend to the linear regression case.
2- The paper then proceeds in extending the WIS-LS idea to the off-policy evaluation problem, notably reformulating the LSTD(lambda) algorithm. Furthermore, a fully online version of the algorithm is provided with constant computational complexity over time. There are two aspects of this part which are not fully clear:
a- In Thm.7 it is shown that WIS-LSTD(lambda) reduces to WIS-LS in the Monte-Carlo, finite-horizon case. Given that WIS-LS corrects only for the output distribution (in this case the sum of rewards), I am not sure I fully understand how this is sufficient in the off-policy case, when the state-action distribution (which corresponds to the input distribution) may vary because of the different policies.
b- The incremental definition of WIS-LSTD(lambda) still requires the inversion of a matrix of the dimensionality of the number of features (m^3), while usual online implementations of TD only have a linear complexity per step. Do you think such a linear-complexity implementation may be actually possible?
3- Finally, the paper reports preliminary empirical results in a simple exact and approximate chain problem. Although the test is very simple, it illustrates the performance of the proposed method along multiple dimensions: number of steps, lambda, and regularizer. Results show a clear advantage of the weight importance sampling solution both in performance and robustness in the parameter selection. The only drawback of this part of the paper is that the problem is really simple and a more extensive empirical evaluation on more challenging problems would definitely benefit to the paper.
4- Although the authors reference most of the relevant literature, no discussion and comparison is provided w.r.t. the recent paper:

Doubly Robust Policy Evaluation and Learning
Authors: John Langford, Lihong Li and Miroslav Dudík
Conference: Proceedings of the 28th International Conference on Machine Learning (ICML-11)

which study a very much related problem. Another very much related paper is

Hirotaka Hachiya, Takayuki Akiyama, Masashi Sugiyama, Jan Peters: Adaptive importance sampling for value function approximation in off-policy reinforcement learning. Neural Networks 22(10): 1399-1410 (2009)

which seems to have an overlap with the current submission.

Clarity
==============

The paper is very well written.

Originality
==============

As reviewed by the authors, the idea of importance sampling has been already applied in off-policy learning in the past. The way the authors bring weighted importance sampling to RL through the clean and elegant formulation of WIS-LS in LSTD is original. Nonetheless, I encourage the authors to have a more thorough discussion with existing literature point out similarities and differences in order to understand better the originality of the contribution.

Significance
==============

Off-policy evaluation is a crucial problem in a huge variety of problems and effective solutions to this problem may have a significant impact.

***************************************** AFTER REBUTTAL

Thanks to the authors for their reply. The comparison sketched in the rebuttal is fair and seems convincing. Thus I confirm my positive impression about the paper and I strongly encourage them to revise the paper including the reference and discussion.
Summary: My overall evaluation of the paper is positive, although I encourage the authors to clarify the points I raised in my review and discuss on how their algorithm compare to the doubly-robust approach and in particular to the adaptive importance sampling solution for off-policy RL.
Author Feedback
Author rebuttal: We thank the reviewers for carefully reading our paper. The main remaining question seems to be how the proposed method compares to the APDIW algorithm proposed by Hachiya, Akiyama, Sugiyama & Peters in 2009. The short answer is that our work for the first time extends weighted importance sampling (WIS), perhaps the best-known and most popular variance reduction technique for off-policy learning, to function approximation and to reinforcement learning with function approximation. This extension is significant in its own right.

That said, it is appropriate to explicitly identify the advantages of our method over APDIW, which was constructed for a similar purpose. In the revision, we would note the following theoretical and practical advantages of our method compared to APDIW:
- The adaptive scheme of APDIW balances a per-decision IS variant and an inconsistent method without importance sampling (named NIW for “no importance weighting”), and is therefore not guaranteed to be consistent. In contrast, the WIS algorithms extend a well-known consistent estimator to parametric function approximation and retain consistency.
- APDIW considers only one-step backups while WIS-LSTD(lambda) considers a full mixture of n-step backups. A proper combination of WIS and eligibility traces is non-trivial and has not been achieved previously.
- From the experiments in Hachiya et al., the empirical performance gain of their algorithm seems modest compared to the several orders of magnitude improvement that we demonstrate for the proper WIS algorithms.

Additionally, for the third reviewer:

First, shifting the input-distribution is not necessary to achieve off—policy learning. In fact, this is not done in most modern off-policy RL methods, including APDIW (Hachiya et al. 2009), GTD, TDC, and GQ(lambda) (Sutton et al. 2009, Maei & Sutton 2010), off-policy LSTD(lambda) (Yu 2010), and the methods of Geist and Scherrer (2014) and Dann et al. (2014). There is sense in this: it is not irrational to desire the accuracy of our value estimates to match the amount of time actually spent in each state under the current behavior, even if this accuracy is measured compared to the value under a different policy. (More formally: the off-policy fixed point is then the minimum of a mean-squared projected Bellman error, where the norm and projection depend on the behavior policy, while the Bellman operator depends on the target policy.) Shifting the input distribution can be interesting, but is an issue orthogonal to those considered in this paper.

Second, the reviewer asks if a linear-complexity implementation of a WIS version of TD may be possible. This is a question of great interest to us and a subject of current work. In short, we believe that it cannot be done exactly, but that an approximation that obtains most of the advantages of WIS is possible with linear per-step computational complexity.

Finally, the reviewer suggests that we discuss the work of Langford, Li & Dudik (2011). This is a good idea, although the relevance of that work is limited because it considers the bandit setting, where the problem of high variance is less pronounced. More importantly, the IS explored in that paper (as a component of their doubly-robust estimation) is conventional IS, not WIS. The ideas of doubly-robust estimation and WIS for function approximation seem to compliment each other rather than to compete. In the revision, we would include a short discussion about that paper to clarify these issues.